# Regulation and Pharmacology of the Cyclic GMP and Nitric Oxide Pathway in Embryonic and Adult Stem Cells

**DOI:** 10.3390/cells13232008

**Published:** 2024-12-05

**Authors:** Alexander Y. Kots, Ka Bian

**Affiliations:** Veteran Affairs Palo Alto Health Care System, US Department of Veteran Affairs, Palo Alto, CA 90304, USA

**Keywords:** nitric oxide, cGMP, guanylyl cyclase, phosphodiesterase, stem cells

## Abstract

This review summarizes recent advances in understanding the role of the nitric oxide (NO) and cyclic GMP (cGMP) pathway in stem cells. The levels of expression of various components of the pathway are changed during the differentiation of pluripotent embryonic stem cells. In undifferentiated stem cells, NO regulates self-renewal and survival predominantly through cGMP-independent mechanisms. Natriuretic peptides influence the growth of undifferentiated stem cells by activating particulate isoforms of guanylyl cyclases in a cGMP-mediated manner. The differentiation, recruitment, survival, migration, and homing of partially differentiated precursor cells of various types are sensitive to regulation by endogenous levels of NO and natriuretic peptides produced by stem cells, within surrounding tissues, and by the application of various pharmacological agents known to influence the cGMP pathway. Numerous drugs and formulations target various components of the cGMP pathway to influence the therapeutic efficacy of stem cell-based therapies. Thus, pharmacological manipulation of the cGMP pathway in stem cells can be potentially used to develop novel strategies in regenerative medicine.

## 1. Introduction

Cyclic guanosine monophosphate (cGMP) is a second messenger cyclic nucleotide synthesized by the family of guanylyl cyclase enzymes and is responsible for the majority of the effects of various endocrine, autocrine, and paracrine mediators, including nitric oxide (NO), natriuretic peptides, and intestinal peptides. NO is a free radical gaseous molecule that is produced in mammals by specific enzymes, NO-synthases (NOS). NO is an important biological messenger and autocrine and paracrine regulator of various signaling processes. NO stimulates the synthesis of cGMP by activating heme-dependent soluble guanylyl cyclase (sGC). Peptide hormones stimulate the production of cGMP by particulate guanylyl cyclases (see [1,2] and references therein).

In general, cGMP is considered a “good” second messenger, and elevation of cGMP levels by various pharmacological agents often produces favorable and protective beneficial effects, especially in cardiovascular diseases. Thus, these positive roles of cGMP and the stimulation of sGC by NO have been extensively used for the development of various drugs that mainly act on the cardiovascular system. The pathway was studied in stem cells to investigate the fundamental biological mechanisms of normal development and to pursue various therapeutic applications of pharmacological agents modulating the components of the pathway (see [1,3] and references therein).

It is essentially impossible to cite every important finding or every original study on the subject directly due to very high number of publications. Thus, in many cases, we referenced recent comprehensive review articles in the text.

The present review provides general description of the cGMP and NO pathways, attempting to minimize the discussion of cGMP-independent aspects of the action of NO, which are only briefly mentioned. Other authors have recently reviewed the roles of NO in stem cell biology in the contexts of mitochondrial function, the delivery of NO using small molecule NO donors, various injectable materials (liposomes, micelles, dendrimers, silica and gold nanoparticles, polymeric particles, and metal–organic frameworks), implantable materials, and various approaches for localized synthesis of NO by enzymes or from prodrugs [4,5,6]. The most important pharmacological agents for manipulating the pathway are then listed. An overview of the role of the pathway in mammalian development and in the biology of embryonic and adult stem cells is presented. We also briefly describe the role of the pathway in cancer stem-like cells. Finally, potentially clinically relevant and mechanistic applications of the pathway in regenerative medicine are listed to emphasize future directions and the use of relevant pharmacological agents for therapeutic purposes.

Thus, we aimed to illustrate the importance of the cGMP and NO pathway as a means for pharmacological manipulation of the features of embryonic and adult stem cells in various diseases to advance various innovative therapeutic approaches.

## 2. General Overview of the cGMP and NO Pathway

A simplified general scheme outlining the molecular mechanisms of the generation and action of cGMP and NO is shown in Figure 1. An overview of the discovery of the pathway has been provided previously (see [1,7]), and recent advances have been summarized (for example, see [8,9,10] and references therein). The majority of the pathway components are presumed to function identically in regular versus stem cells, and specific features of the pathway detected in stem cells are considered to be due to variations in the expression levels of various pathway components.

The intracellular levels of cGMP are regulated by a number of factors targeting its synthesis and degradation. The synthesis of cGMP from GTP is catalyzed by soluble and particulate guanylyl cyclases (GCs) and normally involves the activation of these enzymes by various hormones and mediators. The hydrolysis of cGMP into GMP by phosphodiesterases (PDEs) and the extrusion of cGMP into the extracellular space are the main mechanisms to decrease the levels of cGMP.

### 2.1. NO and sGC

NO is important messaging molecule in almost all biological species and is the main physiological and pharmacological activator of cGMP synthesis by sGC.

In mammals, NO functions as an endocrine, paracrine, or autocrine messenger depending on its source and location. NO is a free radical gas and is thus relatively unstable mainly due to oxidation by O_2_ to nitrite in physiological media in the body, with a short half-life ranging from milliseconds to seconds under aerobic conditions.

Under normal conditions, the majority of NO in the body is produced endogenously by NOS, which are ubiquitous enzymes that utilize amino acid L-arginine as a substrate (for details, see [11] and references therein). The guanidine group of L-arginine is oxidized by oxygen in the heme-containing active site of NOS in a multistage reaction involving several redox cofactors, including FAD/FMH, NADPH, and tetrahydrobiopterin (BH4). There are three major isoforms of NOS: neuronal (nNOS), inducible (iNOS), and endothelial (eNOS). These isoforms are encoded by separate genes. Constitutively expressed NOS isoforms, nNOS and eNOS, are activated by calcium/calmodulin. The expression of iNOS is stimulated by certain cytokines and bacterial endotoxin [11].

In addition to NOS, NO can be generated in various redox reactions of certain nitrogen-containing compounds that are normally consumed or present in the body. These reactions may be spontaneous or are catalyzed by redox enzymes. For example, NO can be generated from nitrite under acidic conditions in the stomach spontaneously, or nitrite can be reduced to NO by bacterial microflora in the oral cavity or gut [12]. Moreover, ambient air usually contains some low levels of NO, which are substantially increased in the case of environmental pollution or contamination. NO is readily interconverted in the body into nitrosonium (NO^+^)-containing species, nitroxyl (NO^−^/NHO), various complexes with transition metals, S-nitrosothiols (RSNO), and other compounds generally known as reactive nitrogen species (RNS) [12].

The paracrine effects of NO are usually a direct consequence of its local diffusion through the plasma membrane between adjacent cells. Autocrine effects are induced by NO produced within the very same cell to regulate diverse functional characteristics. However, NO can also be transported with the blood to reach various remote organs to exert its endocrine effects. These processes involve reversible conversions of NO into certain forms, e.g., S-nitrosylated hemoglobin [13].

Traditionally, the biological effects of NO are considered to be either cGMP-dependent or cGMP-independent. The former are mediated by the main NO receptor sGC. Mammalian sGC is a cytosolic heterodimeric protein containing non-covalently bound heme as a prosthetic group. In general, all tissues express the main sGC subunits, α_1_ and β_1_, to form the ferrous heme-containing heterodimeric α_1_β_1_ sGC. The binding of NO to the heme iron of sGC activates the synthesis of cGMP [14]. The oxidation of the heme into the ferric form during oxidative stress induces the conversion of sGC into heme-deficient apo-enzyme, thus disrupting NO-dependent stimulation of cGMP synthesis [15]. The synthesis of cGMP by apo-sGC can be activated NO-independently by protoporphyrin IX or various pharmacologically active compounds [15].

In addition to sGC, NO interacts with various low and high molecular weight molecules to induce cGMP-independent effects. NO participates in redox signaling and can transform into various reactive nitrogen species (RNS). In direct chemical interactions with certain ROS, NO can scavenge other free radicals, acting as an antioxidant and being consumed as a result of these reactions, depending on its intracellular concentration and compartmentalization [16]. The main targets of direct cGMP-independent effects of NO include protein-bound transition metals and certain specific reactive thiol groups [17].

### 2.2. Peptide Hormones and pGCs

Other pathways of cGMP synthesis are triggered by stimulation of transmembrane GCs by peptide hormones. GC type A (GC-A) is activated by atrial (ANP) and brain natriuretic peptides (BNP) and urodilatin, and GC type B (GC-B) is stimulated predominantly by C-type natriuretic peptide (CNP). Intestinal GC type C (GC-C) is activated by the intestinal peptide hormones guanylin and uroguanylin and by stable toxin a, which is secreted by enterotoxigenic strains of *E. coli* [18].

In oligomeric particulate GCs, the binding of stimulatory peptides to the extracellular region induces allosteric changes in the protein conformation to activate cGMP synthesis by the intracellular catalytic site [18].

Retinal GC isoforms are regulated by calcium and specific adaptor proteins [18].

### 2.3. Control of cGMP Levels

The physiological effects of cGMP are determined by steady-state intracellular levels of this second messenger depending on the rates of cGMP synthesis [18], degradation [19], and extrusion from the cells [20]. The hydrolysis of cGMP is catalyzed by phosphodiesterases (PDEs), a class of enzymes composed of 11 distinct families [19]. PDE isozyme 5 (PDE5) is one of the main PDEs responsible for the degradation of cGMP [19].

The levels of cGMP are very tightly controlled within acceptable limits via multiple redundant feedback mechanisms [21]. Intracellular concentrations of cGMP are normally fluctuating within a narrow range from 0.1 to 1.0 µM. An example of the control components is shown in Figure 2 for a system containing a guanylyl cyclase, cGMP-dependent protein kinase (PKG), and PDE5. This type of regulation is generally able to limit the spikes in intracellular concentrations of cGMP to ensure that an increase is only transient due to the buffering of this second messenger. Long-term changes in the levels of cGMP can be achieved by influencing the expression of the components of the pathway [21].

The biological necessity of this tight regulation can be explained by disastrous consequences of an uncontrolled increase in cGMP, which happens in septic shock in vascular smooth muscle due to excessive activation of sGC by iNOS-produced NO and may result in potentially lethal hypotension [22] or in acute bacterial enteritis in the intestinal epithelium due to excessive activation of GC-C by stable toxin a, leading to acute diarrhea [23,24,25].

### 2.4. Downstream Targets of cGMP

An increase in the levels of cGMP regulates a number of the downstream intracellular effectors, including PKG to induce protein phosphorylation, cGMP-regulated PDEs to regulate the degradation of cGMP and cAMP, and cyclic nucleotide-gated (CNG) ion channels to influence ion transport in the cells [1].

The two main isoforms of PKG encoded by two separate genes, PKGI and PKGII, are differentially expressed in the cells and tissues, thus contributing to the complexity of the pathway [26]. Ultimately, PKG phosphorylation of PKG-specific substrates and common motifs shared with cAMP-dependent protein kinases mediates the regulation of multiple cellular functions, including gene expression, signal transduction, metabolic activity, mitochondrial processes, autophagy, ion transporters and channels, calcium transport in the endoplasmic reticulum, and contractile proteins. Cellular responses can be transient in the case of NO/sGC/cGMP/PKG-dependent smooth muscle relaxation or long lasting in the case of synaptic signaling and memory [26].

Specific crosstalk of the cGMP pathway with cAMP is predominantly mediated by direct activation of PDE2 and inhibition of PDE3 by cGMP [19]. These two PDEs have dual specificity to their substrates and hydrolyze both cGMP and cAMP. Thus, an increase in the levels of cGMP will enhance the hydrolysis of cGMP by PDE2 and PDE3, stimulate cAMP hydrolysis by PDE2, and suppress cAMP hydrolysis by PDE3. The final outcomes are determined by other cell- and tissue-specific factors, including the compartmentalization of relevant proteins and cyclic nucleotides, and relative expression levels of the members of the PKG and PDE families [19].

CNG channels mediate the effects of cGMP and cAMP on membrane potential and calcium transport across the plasma membrane to play a central role in vision and olfaction signaling [27].

Specific combinations of these outcomes ultimately result in certain physiological responses to cGMP depending on the cell and tissue types [28].

## 3. Pharmacological Agents Targeting the cGMP and NO Pathway

Numerous classes of pharmacologically active compounds were developed to target multiple components of the cGMP and NO pathway and have been reviewed in recent publications (see [8,15,29,30] and references therein). These agents have diverse chemical structures and are classified based on their targets. The whole array of the compounds represents a good example of mechanism-based drug development. Some agents have been extensively used in clinical practice for many years.

### 3.1. Direct or Indirect Manipulation of NO Levels

NO gas is relatively unstable and is hard to deliver directly. However, NO inhalation treatment is being successfully used for severe persistent pulmonary hypertension of newborns [31], and some modern approaches for direct NO generation at the site of administration are being developed using various devices, e.g., spark plug-, gliding arc-, and cold plasma-based systems [32,33].

NO donors are the agents that produce NO in the intracellular or extracellular milieu either by spontaneous decomposition or by reactions with intracellular proteins or low molecular weight thiols [34]. Some examples of NO donors include the well-known drugs nitroglycerin, isosorbide nitrate, and sodium nitroprusside. Additional experimental compounds include RSNO that have their own biological effects or release NO enzymatically or in reaction with cysteine [35], diazenium compounds that are spontaneously decomposing to form NO [36], heterocyclic synthetic compounds that form NO in reactions with thiols or transition metals in the cells [37,38], and other types of compounds including hybrid drugs [39]. Recently, various nanoparticle-based delivery systems are being developed to enhance the pharmacokinetics and bioavailability of these agents or even achieve targeted delivery to specific cell types or tissues [40,41].

The synthesis of NO by NOS involves multiple cofactors and is regulated by various signaling pathways [42]. The suppression of NOS activity may be due to the lack of oxygen in hypoxia, excessive production of endogenous inhibitors similar to asymmetric dimethyl arginine [43], or oxidative stress associated with enzyme uncoupling. An indirect increase in endogenous generation of NO can be achieved by stimulation of NOS. Supplementation with L-arginine is expected to provide high concentrations of the main NOS substrate and enhance NO synthesis. The recoupling of uncoupled NOS is achieved by using citrulline or BH4 [42].

The inhibition of the synthesis of NO by NOS is useful for mechanistic approaches to the NO pathways in experimental studies and is usually achieved by treatment with various analogs of L-arginine to lower the NO concentration in the cells indirectly [42].

### 3.2. Regulators of GCs

Stimulators of sGC (BAY 41-2272 and its analogs riociguat, vericiguat, and praliciguat) bind to an allosteric regulatory site of the enzyme to potentiate NO-induced synthesis of cGMP. Riociguat is FDA approved for the treatment of pulmonary hypertension, and approvals for other indications are pending [44].

Activators of sGC act by interacting with the heme-binding region to mimic the conformation of the enzyme with NO-bound heme [44]. These compounds preferentially activate heme-deficient apo-sGC and are thus useful to counteract the disruption of the NO and cGMP pathway caused by oxidative stress with an associated lack of NO and deficiency in heme-containing holo-sGC. The list of agents includes BAY 58-2667 (cinaciguat), ataciguat, and runcaciguat [29].

Inhibitors of sGC activation by NO act by oxidizing ferrous heme iron thus blocking NO binding to the enzyme [45]. Non-specific inhibitors suppress the stimulation of cGMP synthesis by all guanylyl cyclases [25,46].

Recombinant natriuretic peptides and analogs include carperitide (recombinant ANP), ularitide (synthetic urodilatin), nesiritide (recombinant BNP), cenderitide and vosoritide (modified CNP analogs), and other peptides that activate GC-A, GC-B, or both enzymes. The membrane-bound enzyme neprilysin is the main protease responsible for the degradation of natriuretic peptides, and neprilysin inhibitors, e.g., sacubitril, may be used to increase the circulating levels of endogenous ANP and BNP and enhance cGMP production by GC-A [47].

Analogs of intestinal peptide hormones include linaclotide and plecanatide, which are efficient activators of GC-C [48].

### 3.3. Inhibitors of cGMP Degradation by PDEs

Inhibitors of cGMP-specific and dual-specificity PDEs suppress normal degradation of cGMP and thus can increase the intracellular concentrations of this second messenger to enhance the downstream effects of the cGMP pathway [49].

Inhibitors of cGMP-specific PDE5 are widely used in clinical practice for the treatment of cardiovascular diseases, specifically including pulmonary arterial hypertension, and erectile dysfunction, including sildenafil, tadalafil, vardenafil, and avanafil [49].

Some of the effects of inhibitors of PDE3 (milrinone and cilostazol) and PDE1 (vinpocetine) are mediated by suppression of cGMP hydrolysis in addition to cAMP degradation [19].

## 4. The cGMP and NO Pathway in Stem Cells and Development

The first evidence linking cGMP signaling to developmental and regenerative processes was established in the mid-1970s in a study that compared fetal and regenerating liver with normal adult organs to demonstrate a significant decrease in sGC activity and an upregulation of particulate GC [50]. Since then, numerous reports showed various roles of the cGMP pathways in development, stem cells, and regeneration. Availability of diverse pharmacological tools to modulate the cGMP pathways enabled complex manipulations of stem cells in mechanistic studies and therapeutic applications.

GCs and cGMP are considered to have emerged in the evolution of sensory systems in primitive early life forms, including protostome and deuterostome invertebrates. The system then developed into the hormone/messenger receptor cascades [51]. Moreover, various types of NO signaling are highly evolutionarily conserved, and the pathway was suggested to be present in the earliest life forms on Earth [52].

### 4.1. Mammalian Development

The main evidence supporting the pivotal role of the cGMP-NO pathway in early embryonic and subsequent prenatal development comes from the experiments with knockout animal models. The critical role of the pathway in early blastocyst development was initially established in mice by treatment of two-cell embryos in vitro with an inhibitor of NOS, an NO donor, an inhibitor of sGC, and a cell-permeable cGMP analog [53].

Subsequent analysis of developmental abnormalities in various knockout models of single NOS isoforms was not very informative because nNOS, eNOS, and iNOS can compensate for each other due to the paracrine nature of NO and its membrane permeability. Moreover, the lack of endogenous NO is compensated to an extent by environmental NO originating either from the air or from reduction of nitrate and nitrite by various symbiotic bacteria or endogenous redox systems. However, double knockouts and attempts at triple knockouts of all NOS isoforms indicate that NOS-deficient animals manifest low viability, multiple cardiovascular defects, a low number of offspring, nephrogenic diabetes, and arteriosclerosis [54].

Total and non-tissue-specific knockouts of the α1 subunit of sGC develop more or less normally presumably due to compensation by the α2 subunit but still have considerable vascular defects (see [55] for a review and references therein). However, knockouts of the unique β1 subunit of sGC do not develop normally, and 70–80% of the animals die within two days after birth, while the majority of the survivors die within two weeks due to a deficiency in gastrointestinal motility and severe hypertension [56].

A number of developmental defects were reported in mouse knockout models of the genes for natriuretic peptides and their receptors. For example, a recent report demonstrated frequent developmental defects in GC-A^−/−^ knockout mice indicative of abnormal embryonic vascular development [57]. The knockout of the CNP gene causes dwarfism and early death [58], similar to the knockout of one of the main cGMP targets, PKGII [59]. It is possible that parental deficiency in natriuretic peptide signaling may account for these dramatic defects. However, this does not seem to be the case because mice with knocked-down ANP have normal gestation and can produce healthy offspring provided that this offspring expresses normal levels of GC-A [60]. Notably, the knockout of another isoform of cGMP-dependent protein kinase, PKGI, results in various early abnormalities in the cardiovascular and nervous systems in mice [61].

The critical role of the NO-sGC-cGMP pathway in mammalian brain development has been studied in the fetal tissues of gestating rats. Treatment with NOS inhibitors was shown to markedly reduce the differentiation of stem cells into neurons, and this effect was blocked by the co-administration of the PDE5 inhibitor sildenafil. The treatments did not appear to be toxic, since the total number of the cells was unaffected due to an increase in the non-neuronal cell population caused by NOS inhibition, thus confirming the specificity of the effect [62]. In the case of the peripheral nervous system, a study in rats demonstrated that intrauterine exposure to a NOS inhibitor results in the abnormal development of sympathetic innervation and arteries in newborn animals [63].

In addition to a direct role in embryonic development, various NO-related pathways are of particular relevance in the early stages of pregnancy for the maternal organism [64].

### 4.2. Embryonic and Induced Pluripotent Stem Cells

Embryonic stem (ES) cells are a good model for the studies of early development and the differentiation of progenitor cells for various lineages. ES cells are pluripotent and can grow almost indefinitely in culture due to their self-renewal properties. Typically, mammalian ES cells are maintained on a supportive layer of feeder cells, and the cultures have to include some factors to prevent spontaneous differentiation, such as leukemia inhibitor factor for mouse ES cells or basic fibroblast growth factor for human ES cells [65]. To study the differentiation, ES cells are usually plated under low attachment conditions to induce their self-assembly into embryoid bodies (EBs) for several days and to start essentially irreversible general differentiation while limiting self-renewal [65]. Subsequent outgrowth of EBs on a coated matrix produces all types of progenitor cells and some additional differentiated cells depending on specific composition of the media [66].

ES cells have limited therapeutic potential on their own because of high risk of the formation of teratomas in the recipient and the possibility of rejection by the host immune system. However, directed differentiation of ES cells into specific progenitor lineages can offer productive insight into the recruitment of endogenous stem cells in regenerative medicine [67].

Induced pluripotent stem (iPS) cells are very similar to ES cells and are generated by the reprogramming of differentiated somatic cells to express a set of transcription factors that is required for the maintenance of the undifferentiated phenotype [68]. In general, iPS cells and progenitors obtained via partial differentiation of iPS cells have considerably higher therapeutic potential because they can be patient-specific and offer several important advantages over other known types of regenerative therapies [69].

Moreover, iPS cells and certain differentiated cell types derived from them are good models to study various congenital pathologies. For example, the results of transcriptome bioinformatics analysis were used to validate mouse eNOS^−/−^ knockout iPS cell-derived cardiomyocytes as a model for the investigation of molecular mechanisms of congenital heart defects characteristic for intrauterine cardiogenesis [70].

#### 4.2.1. Components of the cGMP and NO Pathway in Embryonic Stem Cells

The general effects of NO on any cell type are determined by the steady-state concentrations of NO and NO-derived molecules, including RNS (see [16] and references therein). The treatment of undifferentiated ES cells with low concentrations of an NO donor protects mouse ES cells from apoptosis induced by the removal of leukemia inhibitory factor, which is required for their normal growth in culture. A similar effect was observed in human ES cells deprived of basic fibroblast growth factor. Moreover, the overexpression of eNOS in mouse ES cells represses the expression of differentiation genes induced by growth factor removal, while treatment with a NOS inhibitor reverses the effects of eNOS overexpression. This finding indicates that the endogenous production of NO in cultured ES cells is important for the self-renewal and survival of ES cell colonies [71] and may regulate the function of the mitochondria and endoplasmic reticulum similar to the effects observed in other cell types [72,73].

However, some NO donors exhibit considerable toxicity in undifferentiated ES cells. For example, the treatment of mouse ES cells with high concentrations of sodium nitroprusside results in apoptosis mediated by the induction of ROS and activation of the mitogen-activated protein kinase pathway and multiple caspases [74]. The treatment of mouse ES cells with high concentrations of an NO donor enhances their differentiation by stimulating the expression of vascular and skeletal muscle-specific markers. Moreover, the transplantation of these NO-treated cells into mice with modeled hind limb ischemia confirmed the commitment of these cells to the mesodermal lineage. The mechanism of these effects of NO may involve the activation of histone deacetylase and a decrease in the acetylation levels of H3 histone, suppressing the expression of pluripotency markers Oct4, Nanog, and Klf-4 [75].

The expression of various components of the pathway was analyzed in cultured mouse and human ES cells, and the changes in the expression of these components were monitored during the differentiation of mouse [76] and human ES cells [77]. The levels of the expression of all isoforms of NOS change during EB-induced differentiation. Specifically, nNOS expression is higher in undifferentiated mammalian ES cells and goes down with differentiation, while the expression levels of iNOS and eNOS are increased upon the differentiation of ES cells. Interestingly, a noncanonical isoform of iNOS, iNOS-2, which is not induced by immune stimulation, unlike iNOS-1, is expressed at this stage, although both isoforms have largely similar functional properties [78]. Moreover, undifferentiated cells do not express sGC because the mRNA levels of the α1, α2, and β1 subunits are very low and insufficient to produce a functional enzyme [55]. Thus, ES cells lack an essential component of the NO-cGMP pathway, and their intracellular levels of cGMP cannot be increased by treatment with NO donors [79].

The lack of the expression of sGC in undifferentiated ES cells may be due to various factors, including the modulation of the steady-state levels of mRNA [80] through its destabilization [81], changes in the promotor activity of the sGC subunit genes [82], redox signaling [83], or epigenetic modifications [84].

All effects of NO in undifferentiated ES cells are very likely to be cGMP-independent, and the only means to increase the intracellular levels of cGMP in these cells are through the activation of particulate GCs with natriuretic peptides. However, the expression of functional sGC is markedly increased during differentiation almost immediately after the start of EB outgrowth cultures [76,79]. Thus, the sGC pathway can be used for pharmacological manipulation of the early stages of differentiation [85].

Additionally, sGC expression, response to NO, and functional properties can be regulated by alternative splicing [86]. Moreover, alternative splicing of sGC appears to be tightly regulated in human ES cells undergoing differentiation, thus possibly contributing to changes in the pharmacological properties of the NO-cGMP pathway [87].

Due to the lack of sGC, an increase in the cGMP concentration in ES cells requires the activation of particulate GCs (GC-A or GC-B). ANP and BNP may have some cGMP-independent effects on stem cells mediated by natriuretic peptide receptor type C (NPR-C), a G protein-coupled receptor that is not coupled to the stimulation of cGMP synthesis. NPR-C is expressed in undifferentiated mouse ES cells, and the stimulation of NPR-C with a specific agonist protects ES cells from apoptosis induced by oxidative stress by blocking the activation of p53 and the suppression of Nanog in response to DNA damage. Moreover, the knockdown of NPR-C by small-interfering RNA (siRNA) results in a dramatic increase in p53 expression and the induction of apoptosis [88].

Recent publications indicate that GC-A plays a specific role in the self-renewal and survival of mouse ES cells. Major evidence is based on alterations in the morphology of cultured ES cell colonies induced by the knockdown of GC-A mRNA; however, it is unclear whether cGMP or PKG are involved in the downstream signaling and whether endogenous levels of natriuretic peptides in the medium or the peptides produced by the cells contribute to GC-A function [89]. The ANP precursor gene is expressed in undifferentiated mouse ES cells, and its expression is decreased during early stages of differentiation. Moreover, at least some of the effects of ANP are suppressed by treatment of the cells with an inhibitor of PKGI [90].

#### 4.2.2. Role of cGMP and NO in Differentiation of Pluripotent ES Cells

The treatment of mouse ES cells with NO donors and overexpression of iNOS significantly increase the number of ES cell-derived cardiomyocytes during the EB outgrowth phase by enforcing the switch toward the cardiac phenotype and inducing apoptosis in the cells that are not committed to cardiac differentiation [91]. The molecular mechanisms of these effects were further investigated in mammalian ES cells [55].

The induction of cardiac lineage-specific differentiation appears to be the signature feature of the NO-sGC-cGMP pathway. The treatment of EB-derived outgrowth cultures with NO donors, activators of sGC, including BAY 41-2272, and their combinations significantly increases the expression levels of cardiac-specific progenitor markers and myocardial proteins [79]. Undifferentiated ES cells should not be sensitive to sGC activation due to the lack of functional sGC expression. However, partially differentiated EB-derived cells start expressing sGC and, thus, can be directed toward the cardiomyocyte lineage. Additional evidence pointing toward a role of cGMP in this process was obtained in ES cells treated with polyphenol curcumin, which induces cardiac-specific differentiation and concomitantly elevates the cGMP content in the cells, suppressing cGMP hydrolysis in the cell-free extract [92].

These observations were subsequently confirmed in a number of independent studies. A detailed investigation of the mechanism demonstrated that the upregulation of PKGI activity during early stages of differentiation mediates the NO-dependent enhancement of mouse ES cell commitment toward the cardiomyocyte precursor lineage [93]. The production of endogenous NO is essential for the commitment of ES cells because NOS inhibitors block this differentiation pathway, while the activation of endogenous NO production by stimulation of AMP-dependent protein kinase and the mTOR (mammalian target of rapamycin) pathway promotes the formation of cardiomyocyte precursors [94]. Similar data were obtained in mouse ES cells treated with ascorbic acid to induce cardiomyocyte-specific differentiation. These effects were associated with an increase in the expression of eNOS and were blocked by an inhibitor of NOS and by scavengers of free radicals [95]. Downstream signaling processes may involve the calcium release pathways through inositol-1,4,5-triphosphate receptors because the knockdown of these receptors with short hairpin RNAs abolishes NO-induced stimulation of cardiomyocyte progenitor marker expression in cultured mouse ES cells [96].

A comprehensive study in mouse stem cells without EB formation demonstrated that their release from pluripotency by the removal of leukemia inhibitory factor results in a subpopulation of cells that expresses various mesodermal markers and eventually serves as a source of cardiac progenitor cells. A unique molecular mechanism appears to involve cGMP-independent S-nitrosylation of histone deacetylase 2 with subsequent release of repression to express mesodermal lineage genes [97].

Interestingly, chemical agents are not the only tools available to influence the differentiation of ES cells. The mechanical stimulation of EBs generated from mouse ES cells promotes vasculogenesis. In this case, the molecular mechanism includes an increase in the synthesis of NO by endogenous NOS, since the formation of the vascular structures induced by a mechanical stimulus is blocked by an inhibitor of NOS [98]. Opposite effects were observed in EBs treated with the antagonist of beta-adrenergic receptors propranolol, which decreases the formation of capillary structures and expression of vascular marker genes. However, the co-treatment of EBs with propranolol and an NO donor restores vasculogenesis, thus suggesting that the differentiation of ES cells into vascular progenitor cells requires the functional NO pathways [99].

The role of endogenous NO produced by NOS in differentiating ES cells is not restricted to the cardiomyocyte lineages. For example, the later stages of the differentiation of mouse ES cells into endothelial precursor cells are significantly inhibited by the treatment of ES cells with a non-specific NOS inhibitor, L-NAME [100]. Subsequent reports indicated that NO-releasing biologically compatible materials are the promising tools for directed differentiation of ES cells. For example, a chitosan-based hydrogel releasing NO in a controllable manner enhances the formation of endothelial progenitor cells (EPCs) in mouse ES cell culture [101].

The treatment of ES cells under specific conditions with an NO donor can promote the differentiation of ES cells toward the pancreatic β-cell phenotype. The effects of NO depend on the modulation of the transcription factors required for the expression of pancreatic β-cell precursor biomarkers and the ability to secret insulin in a glucose-dependent manner [102].

Apparently, the main downstream targets of NO-induced triggers of differentiation are linked to transcriptional activity of the genes encoding for the two major factors involved in the maintenance of pluripotency of ES cells: Nanog and Oct4 [103]. The treatment of mouse ES cells with relatively high concentrations of an NO donor suppresses the transcription of these genes, and similar data were obtained in cultured human ES cells. NO enhances the phosphorylation of p53 repressor protein and thus promotes the binding of phosphorylated p53 and other factors to the promotor region of the *NANOG* gene. Direct relevance of these effects to the levels of cGMP has not been studied in detail. In any case, during the very early stages of differentiation, NO-treated ES cells acquire epithelial morphology and start to express endodermal markers [103].

The studies on differential expression of GC-A and GC-B during embryonic development in the mouse brain indicate that GC-B expression is very high in certain regions of the developing brain and that signaling through CNP and cGMP contributes to the transition of neuronal stem cells to fully differentiated neurons while the expression of GC-A remains at low levels [104]. Various splice forms of ANP precursor mRNA are detected during the differentiation of adipose tissue-derived stem cells and ES cells into cardiomyocytes [105].

An overview of the pathways associated with the effects of cGMP and NO in undifferentiated and early partially differentiated ES cells is shown in Figure 3.

### 4.3. Adult Stem Cells

The modulation of endogenous adult stem cells is one of the main strategies in regenerative medicine for pharmacological interventions to repair damaged cells and tissues. Transplantation is another application area that involves various exogenous allogeneic or autologous approaches, including the expansion of adult stem cells in culture prior to transplantation, lineage-specific differentiation of pluripotent and/or progenitor cells, and reprogramming to produce pluripotent or lineage-restricted stem cells [67]. Mobilization of endogenous stem cells and their recruitment are major factors contributing to long-term recovery from various acute injuries and may be used in combination with transplantation-based approaches.

At least some of the beneficial effects of NO on wound healing are sGC/cGMP/PKG-dependent and involve the enhancement of de-adhesion of epidermal stem cells, providing enhanced recruitment of these cells to the wound sites from their normal niche [106].

Various responses induced by cGMP and NO in adult stem and progenitor cells are expected to increase the effectiveness of tissue engineering and stem/progenitor cell therapy by mesenchymal, hematopoietic, endothelial, neuronal, and skeletal progenitor cells [107]. The present review is predominantly focused on publications that provide direct evidence on the role of various components of the NO and cGMP pathway that can be used as a link to viable therapeutic strategies. Recent advances in the pharmacology of NO enable the development of an array of creative tools for the delivery of NO into stem cells [108].

#### 4.3.1. Cardiovascular Progenitors

NO is known to play important protective roles in the cardiovascular system [109]. Thus, it is reasonable to suggest that at least some of these beneficial effects are mediated by stem and progenitor cells. For example, experimental and clinical data indicate that NO donors and inhibitors of PDE5 promote restorative processes in an ischemic brain after stroke by modulating the regenerative capacity of neuronal stem cells and other cell types involved in angiogenesis, neurogenesis, and oligodendrogenesis to improve the neurological function during subacute recovery from stroke [110]. Moreover, NO is known to boost the general regenerative properties of stem cells during physical exercise, thus benefitting cardiac repair. Overall, NO positively regulates the survival, proliferation, migration, and differentiation of stem cells to directly or indirectly enhance cardiac remodeling and regeneration. This topic has been extensively reviewed by other authors in the context of dietary nitrate and aerobic exercise as a means to mobilize stem cells to promote cardiac repair through stem cell recruitment and proliferation [111].

A comprehensive overview of the general aspects and various applications of regenerative medicine for the treatment of ischemic heart disease by enhancing endogenous cardiac regeneration; the use of mesenchymal stem cells or allogeneic fetal, umbilical, or embryonic stem cells; the autologous transplantation of adipose-, skeletal muscle-, or bone marrow-derived stem cells, iPSCs, or resident cardiac stem cells; and the engineering of myocardial tissue has been provided by other authors [112].

The mechanisms of the cardioprotective and therapeutic effects of NO and cGMP in the context of stem cells have been investigated in a number of preclinical and in vitro studies. For example, a comprehensive investigation of the effects of NO on the differentiation of mouse adult cardiac progenitor cells into mature cardiomyocytes was performed in clonally expanded cultures. Treatment with NO donors increases the expression of cardiac myocyte sarcomeric proteins, and this effect is mimicked by the co-culture of progenitor cells with cardiomyocytes overexpressing eNOS. The effects of NO were blocked by inhibition or knockdown of sGC. Apparently, downstream signaling involves the inhibition of canonical Wnt/beta-catenin signaling, which is essential for cardiac differentiation [113]. This observation is consistent with the critical role of eNOS-derived paracrine NO in the local microenvironment in normal cardiac development and remodeling. A genetic deficiency of eNOS in mice [114] and genetic polymorphism of eNOS linked to lower enzyme activity in human patients [115] are associated with various congenital heart defects. It should be noted that treatment with an NO donor can significantly enhance the survival of human cardiac stem cells and can be potentially used for their preconditioning prior to transplantation [116].

Similar effects were observed in human mesenchymal stem cells isolated from bone marrow and from adipose tissue. The treatment of cultured cells with NO donors for a period of 4 days enhances the expression of cardiac-specific markers and vascular endothelial growth factor, thus suggesting that NO may also increase the pro-angiogenic potential of mesenchymal stem cells [117]. Apparently, endogenous expression of eNOS plays the critical role in these processes, since the overexpression of eNOS enhances the levels of arterial endothelial markers and reduces the expression of venous markers that are mediated by changes in the methylation levels of the corresponding gene promotors [118].

The transplantation of autologous mesenchymal stem cells is one of the preferred modern strategies to repair myocardial damage after acute cardiovascular events. This approach has been used for over 20 years and is characterized by a low incidence of side effects while having relatively low efficacy for actual outcomes [119] despite being tested in multiple clinical trials predominantly due to insufficient numbers or inadequate features of actual stem cells in typical preparations used for these treatments. However, a recent preclinical study suggested intracellular delivery of an NO prodrug metabolized by a mutant β-galactosidase, which is not normally expressed in mammalian cells and was delivered using a lentivirus. These manipulations were able to enhance the overall survival and beneficial paracrine influence of transplanted mesenchymal stem cells in mouse and rat models of myocardial infarction, pointing out potential applications of similar manipulations in a clinical setting, which will require the development of delivery systems suitable for humans for an NO prodrug and its selective enzymatic conversion into NO [120]. A similar approach was used for the modification of mesenchymal stem cells for the treatment of acute kidney injury in a mouse model to enhance the efficacy of the transplanted cells [121]. Notably, a previous study demonstrated an overall less complex but potentially less efficient approach involving the co-transplantation of adipose-derived mouse mesenchymal stem cells with an NO-releasing hydrogel to significantly enhance the efficacy of stem cell therapy in a mouse model of myocardial infarction. Possible mechanisms of these effects can include NO-mediated stimulation of stem cell migration and the secretion of angiogenic factors to attenuate ventricular remodeling [122].

The NO-sGC-cGMP pathway enhances the growth and differentiation of cardiac progenitor cells. In general, the stimulation of cGMP synthesis with natriuretic peptides has similar effects. Natriuretic peptides can enhance the growth of multipotent precursor cells in a neonatal mouse heart and adult mice with myocardial infarction. For example, BNP acts through GC-B to increase the intracellular levels of cGMP and activate PKG. This signaling pathway can contribute to cardiac development and to therapeutic effects of natriuretic peptides in patients with heart failure. In vitro, BNP stimulates the proliferation of cardiomyocyte precursors and enhances their differentiation into cardiomyocytes [123]. However, the stimulation of cGMP synthesis by ANP acting through GC-A seems to have an opposite effect during the later stages of differentiation. Paracrine ANP produced by partially developed mouse ventricles can stunt the growth of cardiac progenitor cells in a cGMP-dependent manner, thus fine-tuning cardiac development through the same regulatory pathway that was used in the early growth of the organ [124]. On the other hand, the injection of newborn and adult mice with BNP increases the number of newly formed cardiomyocytes and proliferating cardiomyocyte precursor cells apparently acting through GC-B and GC-A, respectively, according to experimental data obtained in cultured precursor cells. Thus, application of BNP can be a viable strategy for the stimulation of the regenerative capacity in the heart through the activation of endogenous progenitor cells [125].

An intriguing example of genetic manipulations with stem cells was recently described in a rat model of myocardial infarction. Adipose tissue-derived stem cells were transfected to overexpress eNOS, expanded in culture, and then transplanted into diseased animals. Transfection with eNOS greatly enhances the therapeutic efficacy of the allograft without any adverse effects on the differentiation of the cells into myogenic, neuronal, or endothelial lineages [126].

Evidence emphasizing the protective role of the NO-cGMP pathway was obtained in cardiomyocyte precursor cells derived from mouse ES cells. Ischemic damage was simulated in cultured cells by exposure to hypoxia followed by reoxygenation. Treatment with an NO donor protected the cells in a concentration-dependent manner, while this protective effect was blocked by co-treatment with an inhibitor of PKG. It appears that the effects are sGC-specific because neither NOS inhibitors nor the activation of pGC with BNP had an effect on the cell viability [127]. A similar human model of ES cell-derived cardiomyocytes was used to study the inhibitory effects of NO, cGMP, and PKGI on hypertrophy induced by the treatment of fully differentiated cells with an α-adrenoreceptor agonist. The stimulation of cGMP accumulation in differentiated cardiomyocytes by NO activated cGMP signaling through PKGI and downstream changes in store-operated calcium entry [128], essentially confirming the results obtained in adult cells and thus validating the relevance of this model.

Recent evidence suggests that at least some of the protective effects of mesenchymal stem cells may apparently be due to non-cGMP-dependent mechanisms mediated by iNOS expression via the tumor growth factor-β pathway, which is modulated by an E3 ubiquitin ligase complex [129].

A number of approaches using iPS cells have been recently under development for the treatment of cardiovascular diseases. However, it turned out that despite the extensive manipulation involved in the generation and growth of iPS cells, the cells still inherited metabolic and functional defects of the host organism. For example, iPS cells derived from fibroblasts of obese adult mice were differentiated into endothelial cells that could be used to treat experimental hind limb ischemia; unfortunately, the cells had an impaired capacity to form capillaries as well as lowered migration and proliferation versus iPS-derived endothelial cells from healthy animals. These factors can potentially limit the efficacy of therapy with differentiated iPS cells. Interestingly, the treatment of “obese” cells with pravastatin, an inhibitor of isoprenoid and cholesterol biosynthesis, normalized the functional properties of the cells, and this effect was blocked by an inhibitor of NOS [130]. Thus, NO signaling may play an important role in correcting the undesirable properties of iPS cell-derived therapies.

Treatment with an inhibitor of cGMP phosphodiesterase protects endothelial precursor cells formed during EB phase of mouse ES cell differentiation from damage induced by hyperoxic conditions, while an inhibitor of sGC activation by NO decreases vasculogenesis. Thus, agents that elevate cGMP can rescue endothelial progenitor cells from oxidative stress and preserve vascular repair under various pathological conditions [131].

All these studies clearly support the key role of the NO and cGMP pathway in cardiovascular regenerative approaches involving either endogenous or exogenous stem cell-related therapies.

The main components of beneficial effects of NO and cGMP on cardiovascular regeneration are summarized in Figure 4.

#### 4.3.2. Neuronal Progenitors

Despite extensive evidence supporting the critical role of NO in normal brain functions [132], the mechanisms involved in stem cell differentiation to neurons or other brain cells or their potential applications for stem cell therapy are not clearly defined. However, recent studies in various developmental models point toward an overall stimulatory effect of NO on neuronal stem cells.

In cultured mouse neural progenitor cells derived from embryonic hippocampus, treatment with NO donors increases the proliferation of the cells via the sGC-cGMP-PKG pathway [133]. The effect is biphasic, and early stages of NO-dependent stimulation of cell growth are cGMP-independent, while extended exposure to NO enhances cell growth via an sGC-cGMP-dependent mechanism [134]. The treatment of the cells with high concentrations of NO suppresses their proliferation. Interestingly, the expression of iNOS appears to be important for the stimulation of endogenous cell proliferation in the hippocampi of mice subjected to seizure injury and brain insult [135].

In a model of differentiation of cultured rat neural stem cells, chronic treatment with an nNOS-specific inhibitor significantly reduces the differentiation of stem cells into neurons while increasing the number of progenitor cells. In this case, the mechanisms of NO effects are associated with metabolism of L-arginine and signaling by brain-derived neurotrophic factor [136]. Similar results were recently obtained in cultured neuronal stem cells from mouse embryos. Commitment of the cells to differentiation to neurons is significantly lowered in cells derived from nNOS^−/−^ knockout animals, and the effects can be linked to the changes in histone acetylation induced by histone deacetylase 2 [137].

Human neuronal precursor cells derived from a teratocarcinoma cell line need to migrate out of the colony in order to acquire a full functional neuronal phenotype. It appears that the migration depends on NO signaling because inhibitors of nNOS, sGC, and PKG block this process while the treatment with an NO donor and a cell-permeable cGMP analog enhances the migration [138]. These findings have been confirmed in the cultures of human neuronal progenitor cells, suggesting an important role of the pathway in the development of human brain [139].

NO-stimulated synthesis of cGMP in adipose tissue-derived SCs has been shown to be the main mechanism of valproic acid-induced neuronal differentiation of these cells [140], which may be linked to the effects of histone deacetylase inhibitors that stimulate the expression of sGC [84].

#### 4.3.3. Hematopoietic Progenitors

A study examined the role of NO in the mobilization and engraftment of hematopoietic stem cells using a knockout iNOS^−/−^ mouse model. Apparently, iNOS can be considered an overall negative regulator of the recruitment of hematopoietic stem cells to peripheral blood, and the effects may be mediated by changes in the expression levels of heme oxygenase 1, a known negative regulator of cell migration [141]. This effect can explain the suppression of hematopoiesis associated with various inflammatory processes, and the mechanisms were explored in a recent publication that demonstrated that the critical role of NO in hematopoietic stress is mediated by mitochondrial signaling to drive the proliferation of various stem and progenitor cells during bone marrow regeneration [142].

General impact of NO on hematopoiesis and specific types of hematopoietic stem cells has been extensively reviewed, and the majority of evidence points to multiple roles of eNOS and nNOS in the stimulation of recruitment, mobilization, and quiescence. NO is particularly important for the development and production of megakaryocytes and platelets from precursor cells, regulating the survival of megakaryocytes and their apoptotic pathways, involved in the formation of platelets [143].

#### 4.3.4. Other Progenitors

The role of NO in osteogenesis was investigated in cultured rat bone marrow stromal cells treated with irariin, a potent stimulator of osteogenic differentiation. Irariin induces an increase in the expression of eNOS, iNOS, and PKG and enhances the levels of NO and cGMP in stromal cells. The effects of irariin are significantly suppressed by the co-administration of the inhibitors of NOS and sGC [144]. In another system of cultured human skeletal stem cells, the inhibition of PKGI with the non-specific protein kinase inhibitor H-8 causes their differentiation to osteoblasts while having no effect on the differentiation into osteoclasts or adipocytes. The results were confirmed in an in vivo ectopic bone formation model in mice. The role of PKGI as the main target of H-8 was demonstrated by functional exclusion of alternative protein kinase targets using small interference RNAs [145]. Possible applications of NO in bone regeneration were studied in the cultures of gingiva-derived mesenchymal stem cells treated with NO-releasing synthetic microsphere particles. Delivery of NO slightly enhances cell proliferation and significantly promotes their osteogenic differentiation without adverse effects on cell viability [146].

The NO/sGC/cGMP pathway appears to enhance odontoblastic differentiation of dental papilla cells in a rat model, emphasizing the important role of this pathway in tooth development [147].

The treatment of cultured epidermal stem cells isolated from human skin with an NO donor enhances the migration of the cells, and these findings were confirmed in a mouse superficial partial-thickness burn model. The mechanism apparently involves an increase in the intracellular levels of cGMP and the activation of PKG with subsequent downstream signaling through low molecular weight GTP-binding proteins RhoA and Rac1 [148]. In a similar mouse burn model, we have detected enhanced recruitment of follicle stem cells by treatment of the animals with an NO-generating gel [149].

In a complex in vivo developmental study in mice, the treatment of gestating animals with the NO donor molsidomine enhances myogenic differentiation of fetal endothelial progenitor cells in the embryos. Molsidomine has no effect on the total number of endothelial or hematopoietic progenitor cells but induces some changes in the yolk sac [150].

A recent study demonstrated that adipose tissue-derived stem cells irradiated with non-thermal atmospheric pressure plasma have enhanced proliferation due to NO generated by the plasma source. The treatment had no adverse effects on cell viability or differentiation potential and can be used to accelerate the expansion of similar cells derived from patients and explain enhanced wound healing in tissues treated with this plasma source [151].

### 4.4. Cancer Stem-like Cells

Some cancer cells resemble stem cells in their self-renewal ability. This stem cell-like subpopulation of tumor-initiating cells is characterized by the aberrant expression of ES cell-specific markers of undifferentiated state, including Oct4, Nanog, Sox2, and Klf4. These stem-like cells are defined by enhanced tumorigenic properties, limited abilities to differentiate, and resistance to radiation and chemotherapy. High levels of the expression of iNOS can be one of the factors involved in the maintenance of the cancer stem-like (CSL) phenotype through various mechanisms. Detailed discussion of the roles and therapeutic potential of NO and cGMP in cancer is outside of the scope of this review and has been extensively covered in other publications [152,153,154,155]. Notably, certain unique features of CSL cells can be specifically modulated by cGMP and NO.

For example, serum depletion of cancer cells expressing mutant H-Ras increases the expression of stem cell markers and induces NO synthesis. The treatment of these cells with a NOS inhibitor reverses the induction of stem cell markers and reduces anchorage-independent growth, thus promoting apoptosis [156].

A blockade of endogenous production of NO by iNOS in colon CSL cells by treatment with iNOS inhibitors or by knocking down iNOS expression was shown to reduce tumorigenicity of the cells in vitro and in vivo and to lower the expression of stem cell markers [157]. Molecular mechanisms of these effects may involve cGMP and/or sGC and PKG. Recent in-depth analysis of colorectal CSL cells with highly tumorigenic self-renewing phenotype resistant to apoptosis suggested that various activators of intestinal guanylyl cyclase C, including the peptides linaclotide and plecanatide, may be considered an approach to elevate the intracellular levels of cGMP in CSL cells because of the known antitumor effects of this second messenger molecule [158].

Numerous studied investigated the role of the NO and cGMP pathway in glioma and glioblastoma. An initial report by our group demonstrated that isolated CSL cells from human gliomas lack sGC expression similar to undifferentiated ES cells, and the restoration of the β1 subunit of sGC blocks the aggressiveness of the tumor [159]. A subsequent study demonstrated that at least some of these effects are mediated by direct or indirect transcription factor-like activity of the sGC β1 protein, targeting the promotor of the *TP53* gene to influence the levels of p53 tumor suppressor and thus regulate the cell cycle progression in glioma [160].

In perivascular gliomas driven by the amplification of platelet-derived growth factor, the NO-sGC-cGMP-PKG pathway is activated by NO produced in the tumor-adjacent vascular endothelium to enhance Notch signaling and tumor growth. Moreover, NO increases neurosphere formation in the cultures of these glioma cells. The suppression of NO inhibited Notch signaling in an in vivo mouse model and prolonged the survival of mice, suggesting that the NO/cGMP pathway promotes a perivascular stem-like cellular phenotype in some gliomas and may be a therapeutic target for glioma [161]. These results were partially confirmed using a CSL reporter system based on Oct4 promotor activation [162], and a similar pattern was observed in pancreatic cancer cells selected based on their CSL features [163]. Moreover, glioma CSL cells are responsible for relapse after ionizing radiation therapy. Fractionated irradiation induces the expansion of CSL cells apparently due to the upregulation of iNOS, and inhibition of iNOS reduces the glioma CSL cell population in combination with radiotherapy [164].

The treatment of stem-like cells isolated from the prostate cancer cell line PC3 with inhibitors of PDE5 attenuates various factors contributing to the CSL properties of the cells apparently through the activation of PKG [165]. These findings were confirmed in non-small-cell lung cancer cells (A549 and SK-MES-1), with enhanced expression of stem cell-like features in a 3D system, resulting in epithelial-to-mesenchymal transition and profound chemoresistance. The treatment of these CSL cells with a combination of PDE5 inhibitor tadalafil and cisplatin produced stronger cytotoxic effect than that obtained by treatment with cisplatin alone, indicating that cGMP appears to interfere with the stemness and chemoresistance of CSL cells [166]. Selected CD133^+^/CD44^+^ prostate CSL cells were characterized by upregulation of eNOS expression and overproduction of NO, and the NO-sGC-cGMP-PKG pathway was, in part, responsible for the enhanced growth and androgen resistance of these prostate CSL cells [167].

An increase in the levels of iNOS is characteristic for more aggressive solid tumors. In the case of hepatocellular carcinoma, these aggressive properties are associated with enhanced Notch signaling, which is activated by the iNOS-NO-sGC-PKG pathway in tumor-specific CSL cells with positive expression of CD24 and CD133 [168]. Apparently, iNOS or the downstream signaling components may represent an attractive therapeutic target in this type of cancer.

Several preclinical studies successfully attempted various types of NO-based therapeutic approaches specifically influencing CSL cells to treat certain types of cancer. For example, breast CSL cells have a specific genetic signature, influencing their self-renewal, and this signature is known to be associated with lung metastasis and the NO pathway. As expected, inhibition of the pathway suppresses the migration of CSL cells and may thus be linked with metastatic capacity [169]. On the other hand, myeloid-derived suppressor cells drive the metastatic progression of certain types of cancer and convey CSL properties to regular breast cancer cells, suppressing the activation of antitumor T cells to worsen the survival outcomes in patients. These events are linked to NO-dependent stimulation of Notch signaling [170].

In general agreement with these findings, the treatment of non-small-cell lung cancer cells with NO promoted a CSL phenotype associated with resistance to anoikis, enhanced anchorage-independent growth, and increased migration and invasion. These effects appear to be linked to the activation of caveolin-1 [171]. In epithelial tumor cells, treatment with epidermal growth factor induces CSL features due to the induction of iNOS, which is mediated by the induction of microsomal prostaglandin E synthase-1 [172].

An overview of the effects of NO on SLC cells is presented in Figure 5. Relatively low levels of NO that selectively enhance the formation of cGMP by sGC, if it is expressed, and other agents that enhance the levels of cGMP suppress the CSL cell phenotype in general and thus are expected to suppress tumor growth, metastasis, and recurrence. However, somewhat higher levels of NO produced by iNOS or eNOS to activate the Notch pathway or, in the absence of functional sGC, promote the CSL cell phenotype and thus enhance tumor growth, metastasis, and recurrence due to higher numbers of CSL cells.

In many types of cancer, NO appears to enhance the CSL cell phenotype to promote cancer growth and survival. However, it is also possible to selectively target CSL cells to deliver a very high dose of NO for therapeutic purposes. For example, nanoparticles capable for near infrared light-driven upconversion-dependent release of NO from Roussin’s black salt are preferentially internalized by CSL cells compared to non-CSL cells. A combination of these nanoparticles with traditional chemotherapy was shown to reduce the overall tumorigenic ability and increase drug sensitivity due to the suppression of chemotherapeutic drug efflux from CSL cells [173].

## 5. Regulation of the cGMP and NO Pathway in Therapeutic Applications and Regenerative Medicine

This section of the review attempts to summarize the overall progress in the use of various pharmacological agents influencing the NO and cGMP pathway in regenerative medicine in patients and in various cell culture and animal models. A general overview of the diseases linked to the NO and cGMP pathway and detailed classification of various pharmacological agents has been provided elsewhere [8], and we are focusing our description only on stem cells and regenerative strategies.

Applications of traditional pharmacological agents and formulations summarized in Table 1 demonstrate that almost all known drugs linked to the NO and cGMP pathway have shown predominantly beneficial effects in diverse groups of patients and models of various diseases of the cardiovascular, nervous, and other systems.

Additionally, a number of authors have recently developed very creative approaches to the delivery of specific forms of NO into cells or the environment in various models. These approaches are characterized by rather complex manufacturing processes or advanced manipulations to achieve generally higher levels of NO through diverse means.

Numerous studies used hydrogels to incorporate specific pharmacological agents. The inclusion of unstable S-nitrosoglutathione, an NO donor and sGC activator, in a hydrogel was shown to produce a more stable formulation that enhances cutaneous wound repair and re-epithelization [236] and healing of ischemic wounds in the corresponding rat models [237]. A gelatin–metacrylate hydrogel capable of releasing NO was used to treat diabetic wounds in rats [238]. An injectable hybrid hydrogel dressing containing polydopamine nanosheets was doped with an NO-releasing agent and was able to produce NO on demand for the effective treatment of a full-thickness infected skin defect model in mice [239]. In a mouse model of renal ischemia/reperfusion injury, treatment with NO-releasing nanofibrous hydrogels allowed for enhanced regeneration of the vascular endothelium and accelerated general recovery [240]. Sildenafil citrate has been incorporated into a topical hydrogel that was used to treat traumatic wounds in a rat model [241]. Similar results were obtained in an animal model of radiation wounds, confirming the efficacy of the formulation [242].

Additional unusual vehicles have been manufactured using a silica nanoparticle system capable of NO storage and controlled release in a reversible manner to treat an acute crush nerve injury model in rats to enhance early revascularization and axonal regeneration for improved overall functional recovery [243]. A branched polymer capable of controlled NO release was used for guided bone regeneration mediated by the induction of angiogenesis and osteogenesis in a rat model of critical-sized calvarial bone defects [244].

The coupling of NO to antibacterial peptides is very beneficial for wound healing, since this approach suppresses bacterial growth in the wound while simultaneously stimulating regeneration. The incorporation of these modified peptides capable of NO release into a hydrogel further enhances the formulation [245]. Moreover, a combination of antibacterial and angiogenic properties is very important for the healing of full-thickness infected skin wounds. A photothermal activity-based drug system comprising multiple components was able to release NO and demonstrated better results than the commercially available Aquacel Ag dressing [246].

A unique approach to the delivery of NO in stem cells has been tested by Lee et al. who loaded functional iNOS protein into mineralized nanoparticles formed in the presence of calcium carbonate on an anionic block copolymer template. Interestingly, this presumably invasive approach successfully delivered intact protein into mouse ES cells to increase the intracellular levels of NO and cGMP and eventually promote osteogenic differentiation of loaded ES cells [247].

An overview of recent developments in diabetic wound healing based on controllable NO-releasing biomaterials indicates that these formations compensate for impaired NO synthesis, which is characteristic for diabetes, providing various beneficial effects. However, the translation of these novel technologies to the clinic has a number of challenges due to potential adverse effects [248].

A unique and interesting novel approach involving low-intensity pulsed ultrasound therapy was used to promote recovery from stroke in a mouse model due to enhanced neuro-angiogenesis, which was completely eNOS-dependent because beneficial effects were not observed in eNOS-knockout animals [249].

All these strategies reveal endless possibilities of applications of variable formulations that can selectively target and modulate the components of the NO and cGMP pathway to provide optimal pharmacokinetics and bioavailability to achieve better clinical outcomes.

## 6. Conclusions

Multiple studies clearly support the significant role of the NO and cGMP signaling pathway as one of the critical components in mammalian development. Pharmacological manipulations of the pathways can regulate the intracellular levels of its components and have a substantial impact on undifferentiated pluripotent stem cells. A number of agents and their combinations that target the pathway can be used to influence the differentiation of stem cells, their survival, migration, and homing to the sites of injury. The present review is the first to comprehensively summarize the currently available methods and approaches of manipulating the properties of stem cells within the framework of the cGMP and NO pathway. These advanced tools can be successfully applied at various stages of regenerative therapeutic interventions in diseases of cardiovascular, nervous, and other systems and may have considerable potential in targeting stem-like cells in cancer. Further development of successful strategies can focus on improving the efficiency of the mobilization of endogenous stem cells. The main directions for exogenous stem cell therapy involve beneficial effects of regulators of the NO and cGMP pathway on the expansion, preconditioning, and targeted differentiation of exogenous precursor cells that can be generated from induced pluripotent stem cells.

## Figures and Tables

**Figure 1 cells-13-02008-f001:**
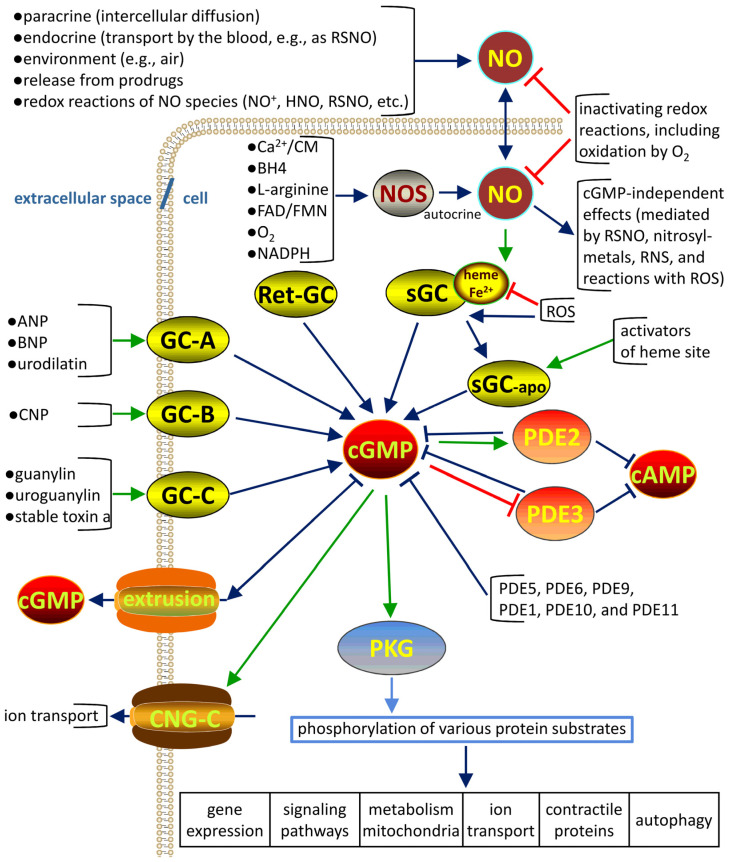
General network of the cGMP and NO pathway. Stimulatory effects are indicated by green arrows; inhibitory effects are indicated by red crossbars; phosphorylation is indicated by a blue arrow; and effects linked to a decrease in the levels of cGMP (or cAMP) are indicated by dark crossbars. Note the cell membrane, which is schematically shown as a lipid bilayer, and the pathways, which include extracellular, intracellular, and transmembrane components, as indicated. Abbreviations: RSNO, S-nitrosothiol; NOS, NO-synthase; CM, calmodulin; BH4, tetrahydrobiopterin; ROS, reactive oxygen species; RNS, reactive nitrogen species; GC, guanylyl cyclase; Ret-GC, retinal GC; sGC, soluble GC; ANP, atrial natriuretic peptide; BNP, brain natriuretic peptide; CNP, C-type natriuretic peptide; GC-A, GC-B, and CG-C are GCs type A, B, and C, respectively; PDE, phosphodiesterase; PKG, cGMP-dependent protein kinase; CNG-C, cyclic nucleotide-gated ion channels.

**Figure 2 cells-13-02008-f002:**
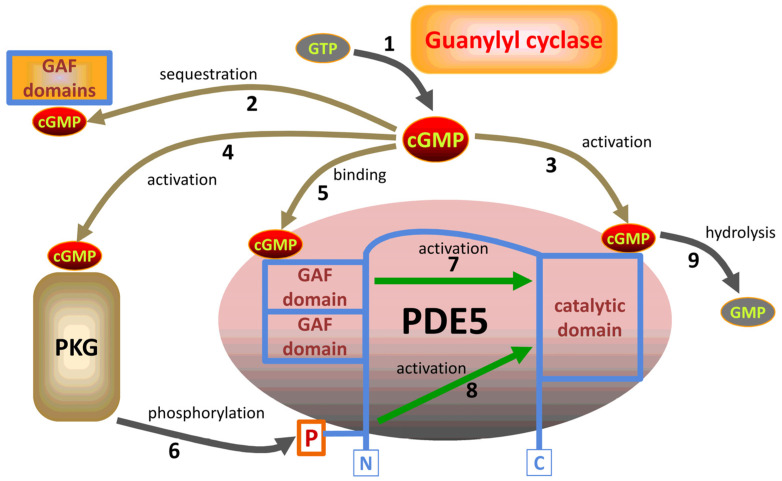
Pathways of feedback regulation of intracellular cGMP contents upon elevation of cGMP concentrations at the level of PDEs. The stimulation of guanylyl cyclases by various agents results in enhanced synthesis of cGMP from GTP (1). Higher levels of cGMP are readily sequestered due to interactions with various cGMP-binding GAF domains (2) of certain mammalian PDEs. Elevated levels of cGMP promote the hydrolysis of cGMP in the active site of cGMP-specific PDE5 due to the changes in enzyme kinetics (3). An increase in cGMP activates PKG (4), which phosphorylates PDE5 (6), and this phosphorylation enhances the catalytic activity of PDE5 via an allosteric mechanism (8). Similar to (3), higher levels of cGMP result in its binding to the GAF domains of PDE5 (5) and enhance the catalytic activity of the enzyme due to allosteric interactions (7). Overall, the mechanisms 3-8 enhance the hydrolysis of cGMP by the catalytic domain of PDE5 (9) to provide tight control over the intracellular contents of this second messenger.

**Figure 3 cells-13-02008-f003:**
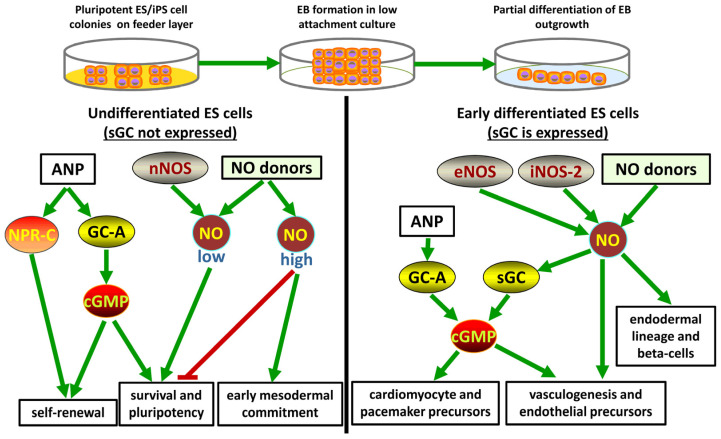
Main effects of the cGMP and NO pathway on undifferentiated and early partially differentiated ES cells. For details, see text. Stimulatory effects are indicated by green arrows, and an inhibitory effect is indicated by a red bar. NO low and NO high correspond to low submicromolar or high micromolar local concentrations of NO. Abbreviations: ES cells, embryonic stem cells; iPS cells, induced pluripotent stem cells; EB, embryoid body; sGC, soluble guanylyl cyclase; ANP, atrial natriuretic peptide; nNOS, neuronal NO-synthase; eNOS, endothelial NO-synthase; iNOS-2, inducible NO-synthase isoform 2; NPR-C, natriuretic peptide receptor type C; GC-A, guanylyl cyclase type A.

**Figure 4 cells-13-02008-f004:**
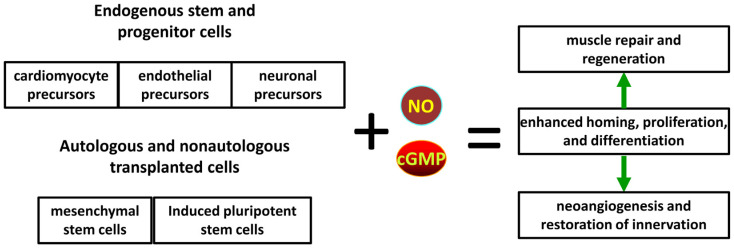
Main components involved in beneficial effects of the cGMP and NO pathway on cardiovascular regeneration and remodeling. For details, see text.

**Figure 5 cells-13-02008-f005:**
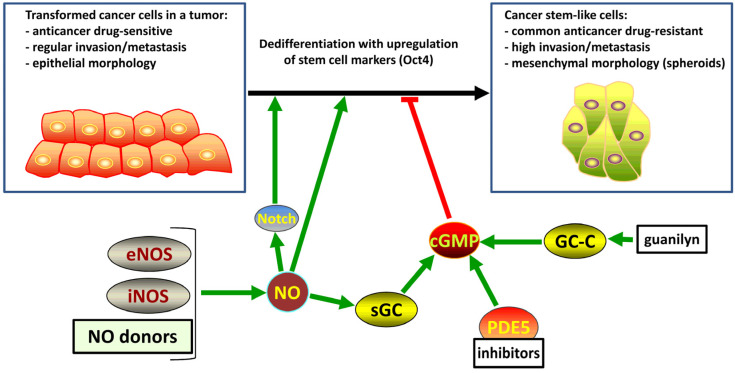
The effects of the NO and cGMP pathway on cancer stem-like cells. Stimulatory effects are indicated by green arrows, and an inhibitory effect is indicated by a red bar. Abbreviations: eNOS, endothelial NO-synthase; iNOS, inducible NO-synthase; sGC, soluble guanylyl cyclase; GC-C, guanylyl cyclase type C; PDE5, phosphodiesterase 5.

**Table 1 cells-13-02008-t001:** Use of pharmacological agents targeting the NO and cGMP pathway in regenerative biomedical and clinical studies.

Agent	Model	Effect	Reference
** *Organic nitrates as NO donors and sGC activators* **			
Nitroglycerin	Cosmetic filler implant-induced necrosis during soft tissue augmentation in patients	Beneficial treatment	[174]
	Human bone marrow-derived MSCs in culture	Increased proliferation and osteoblastic differentiation	[155]
Isosorbide dinitrate	Model of shoulder rotator cuff injury in cultured patient biopsies of supraspinatus and ipsilateral deltoid	Promoted growth to amplify atrophic skeletal muscle regeneration	[175]
	Cultured MSCs	Attenuated high glucose-induced senescence	[176]
	Satellite stem-like cells in skeletal muscle in a zebrafish model	Activated growth or regeneration	[177]
	Age-related sarcopenia in mice in quadriceps muscle	In combination with exercise, increased muscle mass and stimulated cell proliferation to enhance sarcolemmal integrity and vascular density via the activation of muscle stem cells	[178]
	Mouse model of muscular dystrophy induced by knockdown of alpha-sarcoglycan	Combination with ibuprofen stimulated regenerative capacity of myogenic progenitor cells to maintain voluntary movement and exercise resistance	[179]
	General atrophy and sarcopenia in an isolated aging muscle model	In combination with stretching, rescued age-related refractory resistance of satellite cells to proliferation, compensating for a decrease in NO production	[180]
Pentaerythritol tetranitrate	Rodent in vivo and human cell culture models	Increased the number of circulating EPCs and their incorporation into vascular structures	[181]
	Patients with symptomatic coronary artery disease validated by coronary angiography	Significantly increased the number of circulating CD34^+^ EPCs and enhanced the colony-forming ability of EPCs isolated from patients without influencing endothelial function	[182]
** *Various NO donors* **			
Molsidomine (SIN-1)	Cell culture model of myoblast myogenesis	Increased myofiber area to stimulate myoblast differentiation without influencing proliferation or migration of myoblasts	[183]
	Muscle dystrophy and cardiotoxin-induced repetitive acute and chronic damage in cell culture and in alpha-sarcoglycan knockout mice in vivo	Stimulated the proliferation of satellite cells in a cGMP-independent manner to delay a reduction in the satellite cell pool to enhance muscle regeneration	[184]
	Mouse ES cell-derived neuronal stem and precursor cells	Enhanced the proliferation	[185]
	Embryonic and fetal myogenesis in mice in vivo	Enhanced myogenesis	[150]
Sodium nitroprusside	Isolated lineage-negative mouse hematopoietic stem cells suitable for transplantation	Increased the number of CD34^+^ cells due to an increase in CD34 expression to improve the engraftment of juvenile stem cells while decreasing the engraftment of adult stem cells	[186]
	Cultured human periodontal ligament stem cells	Promoted osteogenic differentiation and reduced adipogenic differentiation	[187]
Nicorandil	Rat model of bilateral renal ischemia/reperfusion injury	Combination of the drug with drug-pretreated MSCs enhanced MSC survival and proliferation to alleviate pancreatic insufficiency	[188]
	Rat model of isoproterenol-induced heart failure	In combination with MSC transplantation alleviated cardiac hypertrophy, fibrosis, and inflammation by increasing angiogenesis and MSC homing	[189]
	Animal model of ischemic stroke	Enhanced survival of transplanted MSCs preconditioned with the drug	[190,191,192]
Nitrite/nitrate	Mouse and rabbit animal models of vascular graft prosthetics for vascular bypass surgery	Promoted vascular regeneration while attenuating intimal hyperplasia and calcification	[193]
** *NO-synthase cofactors* **			
Tetrahydrobiopterin	Overview of multiple cofactor deficiency models	Deficiency decreased NO production and impaired the mobilization and function of EPCs in diabetes and other diseases	[194,195]
	Mouse model of salt-sensitive hypertension	Protection of EPCs that is only partially mediated by NO	[196]
	Cultured human EPCs	Enhanced biosynthesis of the cofactor enhanced the regenerative capacity of the cells	[197]
	Cultured EPC subset isolated from patients with coronary artery disease	Deficiency of the cofactor was associated with impaired functional properties of the cells	[198]
** *Natriuretic peptides* **			
Atrial natriuretic peptide	Cultured erythroid progenitor cells	Directly stimulated erythroid colony formation	[199]
Brain natriuretic peptide	Mouse model of myocardial infarction	Directly stimulated the proliferation of resident endothelial cells but did not influence their differentiation to enhance vascularization	[200]
	Mouse model of myocardial infarction	Stimulated the proliferation of endogenous cardiac progenitor cells and their differentiation into cardiomyocytes to increase cardiac contractility and decrease remodeling	[125]
	Neonatal mice	Enhanced the proliferation of hematopoietic/multipotent stem cells expressing stem cell antigen-1 in the heart	[123]
C-type natriuretic peptide	Mouse model of somatosensory neuron regeneration	Deficiency in the hormone or guanylyl cyclase type B impaired heat sensing and nociception but did not influence motor coordination	[201]
** *Phosphodiesterase 3 blockers (both cGMP and cAMP effects)* **			
Cilostazol	Rat model of olfactory bulb hypoperfusion	Increased the number of neuroblasts and enhanced their survival and differentiation to enhance neurogenesis	[202]
	Rat model of monocrotaline-induced pulmonary hypertension	Combination of the drug with bone marrow-derived EPCs had a strong beneficial effect	[203]
	Rat and mouse models of cerebral hypoperfusion	Increased survival of oligodendrocyte progenitor cells, promoting the restoration of the white matter and recovery of cognitive decline	[204,205]
	Mouse model of transient forebrain ischemia	Reduced pyramidal cell loss and increased the number of bone marrow-derived EPCs to enhance neovascularization in the hippocampus	[206,207]
	Rat model of carotid balloon injury	Enhanced re-endothelization by stimulating the adhesion, migration, and proliferation of bone marrow-derived EPCs	[208]
** *Phosphodiesterase 5 inhibitors (mainly cGMP effects)* **			
Vardenafil	People with various cardiovascular risk factors with reduced number of circulating bone marrow-derived EPCs	Increased the number of EPCs	[209]
	Patients with erectile dysfunction and various types of carotid lesions	Increased the number of circulating EPCs	[210]
	Mouse model of unilateral hindlimb ischemia	Enhanced mobilization of EPCs in bone marrow and peripheral blood contributing to neovascularization and blood flow recovery	[211]
Tadalafil	Patients with erectile dysfunction and cultured cells	Chronic treatment increased the number of circulating EPCs to enhance flow-mediated dilation in the brachial artery	[212,213,214]
	Rat model of acute myocardial infarction and cultured cells	Enhanced therapeutic effect of MSC transplantation, with enhanced neomyogenesis in the infarct and peri-infarct regions and attenuated remodeling	[215,216]
	Patients with erectile dysfunction and metabolic syndrome	Increased the fraction of EPCs (CD45^−^/CD34^+^/CD144^+^)	[217]
	Rat model of myocardial infarction	Enhanced the survival of MSCs and therapeutic effect of MSC administration, including cardiac function and blood vessel density	[218]
	Cultured rat MSCs	Enhanced the survival of the cells after hypoxia/reoxygenation	[219]
	Rat model of cavernous nerve injury	Combination of the drug with bone marrow-derived MSCs restored nitrergic relaxation and completely recovered erectile function	[220]
Sildenafil	Rat model of focal cerebral ischemia induced by occlusion of the middle cerebral artery and embolic stroke and cultured neurospheres	Enhanced neurogenesis and functional recovery through the cGMP pathway	[221]
	Mating male mice	Suppressed fertilization and impaired early embryo cleavage	[222]
	Patients with idiopathic pulmonary hypertension and with Eisenmenger syndrome	Dose-dependently increased the number of EPCs, contributing to general positive effects of the drug	[223,224]
	Aging mice in a model of focal cerebral ischemia and cultured stem cells	Induced the amplification of neural stem cells and promoted neuronal differentiation	[62,225]
	Rat model of dilated cardiomyopathy	Combination with autologous adipose-derived MSCs enhanced angiogenesis and increased left ventricular ejection fraction	[226]
	Rat model of monocrotaline-induced pulmonary hypertension	Combination with bone marrow-derived EPCs reduced the recruitment of c-kit-positive progenitor cells and decreased pulmonary remodeling	[227,228]
	Mouse model of myocardial infarction and cultured MSCs	Combination with adipose-derived MSCs, but not skeletal muscle-derived stem cells, improved the survival of MSCs after hypoxia/reoxygenation	[229,230]
	Aging rat model of erectile dysfunction induced by bilateral cavernosal nerve resection	Combination with muscle-derived stem cells failed to benefit the animals	[231]
	Cultured mouse neural stem cells	Enhanced proliferation	[232]
	Various models of stem cell therapy of erectile dysfunction and ongoing clinical trials	Enhanced allograft and homing and recruitment of endogenous stem cells	[233]
	Cultured skeletal myoblasts	Enhanced the proliferation by increasing calcium availability	[234]
	Cultured rat oligodendrocyte precursor cells	Suppressed internal oligodendroglial differentiation	[235]

Abbreviations: sGC, soluble guanylyl cyclase; MSCs, mesenchymal stem cells; EPCs, endothelial progenitor cells.

## Data Availability

Not applicable.

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
