# Peer review of "Regulation and Pharmacology of the Cyclic GMP and Nitric Oxide Pathway in Embryonic and Adult Stem Cells"

_cells, 2024, doi:10.3390/cells13232008_

Round 1

Reviewer 1 Report

Comments and Suggestions for Authors

The present manuscript is an exhaustive review on snthesis and regulation of the second messenger cGMP in different cell types and in different stages of organ differentiation. Tissue levels of cGMP are regulated by numerous endogeneous factors, such as NO (activating sGC) or peptides (activating membrane bound GCs). The authors describe the metabolic regulating in a comprehensive form. However, the plenty of information is little difficult for a reader not so familiar with this topic.

Different approaches may influence cGMP levels, thus affecting important metabolic and developmental functions.

Particularly, the authors focus on the role of cGMP and its regulation in stem cells fo different grade of specialization.

This excellent manuscript can by published in the present version.

Author Response

Q1-1. The authors describe the metabolic regulating in a comprehensive form. However, the plenty of information is little difficult for a reader not so familiar with this topic.

A1-1. We did not want to reduce the overall amount of information. To somewhat improve readability of the text for a reader who is not very familiar with the topic of the review, we have included a brief summary after each section.

Reviewer 2 Report

Comments and Suggestions for Authors

The manuscript entitled Regulation and pharmacology of the cyclic GMP and nitric oxide pathway in embryonic and adult stem cells, brings a comprehensive approaching of NO pathway and pharmacologic manipulation in various diseases.

Observations

Introduction

Please indicate references for every statement in paragraph 1 and 2. (line 24-40).

- line 41-47 - please move this paragraph in material and methods chapter. 

Please provide the aim of this study more accurate. The last paragraph contains only the description of the main chapters of your manuscript.

Fig 1 - you need to mention there are intracellular and extracellular processes, and to indicate the cell membrane in the legend. 

- line 93-94 - please add ref.

- line 95-97 - please add ref

There are overall missing references - line 106, 125, 139, 141, 144, 151, 158, 160, 183, 192, 194, 197, 232, 236, 244, 246, 248, 265, 334

- line 217 - give some examples of such devices

- line 356 - you need to write in the title ES without abbreviation.

The next paragraph has to many sentences without references, and only 2 ref at the end of paragraph.

- line 407-415 - only one ref.

- line 416 - a number of publications and only one ref??

- line 486-495 - only one ref

- same for the next paragraph

- line 643-649 - please add ref.

The Conclusions chapter is to general, and sounds like an abstract. Please focus  on the main ideas on your manuscript, on the novelty of your work, and on contribution to this research field.

Author Response

Q2-1. Please indicate references for every statement in paragraph 1 and 2. (line 24-40).

A2-1. Very general topics briefly mentioned in this part of the manuscript are considered textbook materials and might not require specific references. We decided to cite Dr. Murad’s Shattuck lecture overview as the main source of information. There are hundreds, if not thousands, of general reviews by other authors that can be also cited in this section. Additionally, we selected two more recent mega-reviews on NO and cGMP (references 2 and 3, respectively, in the revised version of the text). Please note that all changes based on the comments of the reviewers are marked in yellow highlight in the revised version, and line numbering refers to the submitted revised pdf file.

Q2-2. - line 41-47 - please move this paragraph in material and methods chapter.

A2-2. Since our manuscript is not an experimental study but is a review, we are not supposed to include a Materials and Method section. Our review is not a meta-analysis or very systematic bibliographic analysis of the literature. Thus, we think that inclusion of a Methods section is not appropriate. However, we included this text into a small new subsection in the Introduction section of the manuscript (1.1. Approach to literature analysis) starting from line 41 in the revised pdf file.

Q2-3. Please provide the aim of this study more accurate. The last paragraph contains only the description of the main chapters of your manuscript.

A2-3. We have added the final paragraph to the Introduction section to better reflect the aims of the review on lines 63-65 of the revised pdf file as follows:

“Thus, we aimed to illustrate the importance of the cGMP and NO pathways as a means for pharmacological manipulations of the features of embryonic and adult stem cells in various diseases to advance various innovative therapeutic approaches.

Q2-4. Fig 1 - you need to mention there are intracellular and extracellular processes, and to indicate the cell membrane in the legend.

A2-4. The legend to Figure 1 has been edited as requested by inclusion of the following sentence in the revised version of the text:

“Note the cell membrane, which is schematically shown as a lipid bilayer, and that the pathways include extracellular, intracellular, and transmembrane components, as indicated.”

Q2-5. - line 93-94 - please add ref.

- line 95-97 - please add ref

There are overall missing references - line 106, 125, 139, 141, 144, 151, 158, 160, 183, 192, 194, 197, 232, 236, 244, 246, 248, 265, 334

A2-5. The references have already been present in the initial version at the end of the corresponding paragraphs or statements, which comprise several sentences or whole paragraphs. Note that references cited in these contexts are representative comprehensive reviews, which include multiple references to the original publications. We have revised the placements of the references to resolve potential ambiguity as requested.

Q2-6. - line 217 - give some examples of such devices

A2-6. We have listed spark plug-, gliding arc-, and cold plasma-based systems as examples on line 223 in the revised pdf file.

Q2-7. - line 356 - you need to write in the title ES without abbreviation.

A2-7. The line has been corrected as requested.

Q2-8. The next paragraph has to many sentences without references, and only 2 ref at the end of paragraph.

A2-8. This was done intentionally because the contents of the paragraph are fully covered by three references cited at the beginning and at the end of the paragraph.

Q2-9. - line 407-415 - only one ref.

- line 486-495 - only one ref

- same for the next paragraph

- line 643-649 - please add ref.

A2-9. The contents in all these passages are fully reflected in cited references Please note that we have very carefully checked the text to exclude potential ambiguity and added additional mentions of the citations already present in the text for clarity.

Q2-10. - line 416 - a number of publications and only one ref??

A2-10. There are a total of 4 publications directly relevant to the subject, and two of these are cited. For clarity, we have replaced “A number of recent publications” with “Recent publications” on line 424 in the revised pdf file.

Q2-11. The Conclusions chapter is to general, and sounds like an abstract. Please focus  on the main ideas on your manuscript, on the novelty of your work, and on contribution to this research field.

A2-11. We decided to keep the Conclusions section very concise. Thus, we have not significantly expanded this section, while trying our best to change the wording (lines 937-942 of the revised pdf file) to reflect your suggestions.

Reviewer 3 Report

Comments and Suggestions for Authors

This is a very timely, nicely written and extremely comprehensive review in the field of NO signalling and its role in stem cell regulation. I liked a lot the way how the authors have summarised and discussed the available literature, and this work will be definitely of interest for the broad research community interested in this topic. I have just some minor points which may need improvement:

1. Page 5 line 150. PDE5 is referred to as an Isoform. correct would be to talk about the PDE5 family.

2. I like the way how the authors have introduced most relevant drug developments in the filed of cGMP elevating drugs. Two more things here could be added to the section 3.2 on page 7 - vosoritide and the name of neprilysin inhibitor which is sacubitril.

3. line 263 onwards in the section 3.2 on page 7 - this section should be 3.3 I guess. PAH can be added as another important indication of PDE5 inhibitors.

4. The first para on page 15 described the use of ES derived myocytes as a model to show protective effects of cGMP elevating drugs agains prohypertrophic stimuli which is also the case in adult myocytes. Since this effect is not directly related to the regulation of differentiation /profiferation, it can be probably omitted. 

Author Response

Q3-1. Page 5 line 150. PDE5 is referred to as an Isoform. correct would be to talk about the PDE5 family.

A3-1. We agree that traditional “isoform” terminology with regard to PDEs is largely outdate and used predominantly in the older literature. Thus, we have replaced “isoform” with “isozyme” or “members of the PDE family” in appropriate contexts (lines 156 and 206, respectively, in the revised pdf file).

Q3-2. Two more things here could be added to the section 3.2 on page 7 - vosoritide and the name of neprilysin inhibitor which is sacubitril.

A3-2. Vosoritide was added on line 263 in the revised pdf file, and sacubitril was added on line 266, as requested. Unfortunately, we have been unable to find references describing the use of these two agents in the context of stem cells.

Q3-3. line 263 onwards in the section 3.2 on page 7 - this section should be 3.3 I guess. PAH can be added as another important indication of PDE5 inhibitors.

A3-3. The subheading numbering has been corrected, and PAH has been added as requested on line 275 in the revised pdf file.

Q3-4. The first para on page 15 described the use of ES derived myocytes as a model to show protective effects of cGMP elevating drugs agains prohypertrophic stimuli which is also the case in adult myocytes. Since this effect is not directly related to the regulation of differentiation /profiferation, it can be probably omitted.

A3-4. The reference number 125 has been included mainly to illustrate the usefulness of ES cell-derived cardiomyocytes for studies of molecular mechanisms of hypertrophy. Unfortunately, we have not clearly worded the description of the original work. We have thus revised the phrasing on lines 643-647 in the revised pdf file to better reflect the intended meaning as follows:

“A similar human model of ES cells-derived cardiomyocytes was used to study the inhibitory effects of NO, cGMP, and PKGI on hypertrophy induced by treatment of fully differentiated cells with an α-adrenoreceptor agonist. Stimulation of cGMP accumulation in differentiated cardiomyocytes by NO activated cGMP signaling through PKGI and downstream changes in store-operated calcium entry [125], essentially confirming the results obtained in adult cells and thus validating the relevance of this model.”